# The Estimation of Grassland Aboveground Biomass and Analysis of Its Response to Climatic Factors Using a Random Forest Algorithm in Xinjiang, China

**DOI:** 10.3390/plants13040548

**Published:** 2024-02-17

**Authors:** Ping Dong, Changqing Jing, Gongxin Wang, Yuqing Shao, Yingzhi Gao

**Affiliations:** 1College of Grassland Science, Xinjiang Agricultural University, Urumqi 830052, China; 320212479@xjau.edu.cn (P.D.); wanggongxin24@163.com (G.W.); gaoyz108@nenu.edu.cn (Y.G.); 2Key Laboratory of Grassland Resources and Ecology of Xinjiang Uygur Autonomous Region, Urumqi 830052, China; 3Key Laboratory of Grassland Resources and Ecology of Western Arid Desert Area of Ministry of Education, Urumqi 830052, China; 4College of Resources and Environment, Xinjiang Agricultural University, Urumqi 830052, China; 17836067476@163.com

**Keywords:** random forest, grassland type, aboveground biomass, climatic factors

## Abstract

Aboveground biomass (AGB) is a key indicator of the physiological status and productivity of grasslands, and its accurate estimation is essential for understanding regional carbon cycles. In this study, we developed a suitable AGB model for grasslands in Xinjiang based on the random forest algorithm, using AGB observation data, remote sensing vegetation indices, and meteorological data. We estimated the grassland AGB from 2000 to 2022, analyzed its spatiotemporal changes, and explored its response to climatic factors. The results showed that (1) the model was reliable (R^2^ = 0.55, RMSE = 64.33 g·m^−2^) and accurately estimated the AGB of grassland in Xinjiang; (2) the spatial distribution of grassland AGB in Xinjiang showed high levels in the northwest and low values in the southeast. AGB showed a growing trend in most areas, with a share of 61.19%. Among these areas, lowland meadows showed the fastest growth, with an average annual increment of 0.65 g·m^−2^·a^−1^; and (3) Xinjiang’s climate exhibited characteristics of warm humidification, and grassland AGB showed a higher correlation with precipitation than temperature. Developing remote sensing models based on random forest algorithms proves an effective approach for estimating AGB, providing fundamental data for maintaining the balance between grass and livestock and for the sustainable use and conservation of grassland resources in Xinjiang, China.

## 1. Introduction

Grasslands are one of the most widespread terrestrial ecosystems globally and play a crucial role in global carbon cycling, climate regulation, and water conservation [1,2]. The aboveground biomass (AGB) of grassland is a crucial indicator for evaluating vegetation use and monitoring grassland ecosystems. Efficient and accurate estimation of AGB in natural grasslands is crucial for understanding their carbon source and sink capacities [3,4]. The rapid development of pastoral economies and global warming have led to severe degradation of grasslands and a decline in ecosystem stability, ultimately affecting the carbon sequestration capacity [5]. There are abundant grassland resources in Xinjiang, China, which have high ecological and economic value [6,7]. Utilizing remote sensing technologies with high spatial and temporal resolutions and efficient machine learning algorithms is a valuable approach for estimating grassland biomass, enabling the monitoring of grassland growth and managing the balance between grass and livestock [8,9]. Therefore, studying the patterns of grassland change and its meteorological factors in the context of global warming and environmental degradation is crucial for protecting and improving natural grasslands in Xinjiang [10,11].

The estimation of grassland biomass can be improved using various methods. Traditional harvesting methods with low estimation accuracy are time-consuming, making it challenging to predict biomass on a large scale. Therefore, remote sensing data with high spatial and temporal resolutions can improve the inversion accuracy of models [12,13]. High-resolution SPOT and Landsat satellite data are suitable for small-scale estimations [14,15]. There are many options for the spatiotemporal resolution of MODIS remote sensing data that can be used to estimate grassland AGB in different regions [16,17,18]. Additionally, AGB estimation is mostly based on the linear regression of univariate vegetation indices [19,20], and there is some level of uncertainty in the estimation using a single factor. The estimation of AGB is also influenced by various factors such as climate change, human activities, and geography. However, most studies have not fully considered the effects of different variables [21]. Studies have shown that the warming and humidification trends caused by climate change make the ecological environment more suitable for promoting the growth of local grassland biomass [22]. Temperature changes affect various aspects, such as the start date and duration of grass-growing seasons, whereas precipitation impacts vegetation cover and grassland biomass [23].

In recent years, the use of remote sensing data to upgrade measurements to the regional scale and apply machine learning methods for AGB estimation has become popular [24]. Machine learning algorithms can handle nonlinear relationships and complex data structures and can automatically select feature variables and optimize parameters to improve the accuracy and generalization of grassland biomass prediction [25]. The commonly used machine learning algorithms for estimating aboveground biomass include random forest, support vector machine, neural networks, and regression analysis, among others. These algorithms are robust and enhance the generalizability of the grassland AGB model compared to traditional linear regression. For instance, the AGB model based on the random forest algorithm developed by Yu et al. using MODIS VIs, topographic, and climatic factors outperformed the single NDVI regression model for grassland AGB estimation on the Qinghai–Tibet Plateau [26]. Compared to bagging, boost, and support vector machine models, the AGB model using the random forest algorithm obtained higher values for the mean Pearson coefficient and the symmetric index of agreement on the Loess Plateau [27]. Additionally, some studies compared the effectiveness of various machine learning algorithms for modeling grassland biomass and found that the model accuracy of the AGB model using the random forest algorithm was more reliable [28,29]. It provides stronger generalization capability than traditional linear regression and achieves higher accuracy and reliability in different regions and conditions, making it become a very promising estimation method. In addition, previous studies have combined multiple variables to develop high-precision AGB models using random forest algorithms for the Loess Plateau and the Sanjiangyuan area [27,30]. However, there have been only limited studies on the prediction of grassland AGB and its spatiotemporal variations in different types of grasslands in Xinjiang at a regional scale. Due to Xinjiang’s vast grasslands, diverse topographical features, and unique climate, the distribution of grasslands is highly uneven across the region. Therefore, it is necessary to conduct an analysis based on different types of grasslands.

In this study, we created an estimation model for grassland AGB in Xinjiang by combining ground-truthed grassland AGB data, considering climate, vegetation index, and other factors, and explored the spatial and temporal variations in grassland AGB and its relationship with climatic factors. This study aimed to (1) establish a suitable AGB model for the Xinjiang grasslands based on the random forest algorithm; (2) analyze the spatial and temporal changes in grassland AGB in Xinjiang from 2000 to 2022; and (3) explore the impact of climate change on the spatiotemporal dynamics of grassland AGB in Xinjiang. The results can provide technical support for accurately estimating grassland AGB, understanding the status of grassland resources, and informing the sustainable development of grasslands in Xinjiang.

## 2. Results

### 2.1. Model Establishment and Accuracy Evaluation

Figure 1 shows the correlation between the modeling variables and grassland AGB in Xinjiang. The correlation of environmental variables with grassland AGB, from highest to lowest, were EVI (0.48) > NDVI (0.45) > LAI (0.42) > ET (0.36) > GPP (0.35) > PRE (0.08) > TEMP (−0.07). Among these parameters, EVI exhibits the highest correlation with AGB (R = 0.48, *p* < 0.01).

This study aimed to evaluate the accuracy of the grassland AGB estimation model. A total of 20% of the sample data was used to validate the accuracy, and a scatter plot was created to compare the estimated AGB from the model with the measured AGB in the field. In the test set, the estimated AGB values were concentrated between 0 and 200 g·m^−2^, with a coefficient of determination (R^2^) and the root mean square error (RMSE) of 0.55 and 64.33 g·m^−2^, respectively (Figure 2a). The AGB model tended to underestimate some high AGB values and overestimate some low values. However, there was a good linear relationship between the estimated and measured AGB values. When the measured grassland AGB was below 100 g·m^−2^, most of the estimated AGB values were higher than the measured values, while for measurements above 100 g·m^−2^, the estimated values tended to be lower. Figure 2b shows that the estimated grassland AGB and measured grassland AGB have nearly equal means of 85.33 g·m^−2^ and 85.39 g·m^−2^, respectively, with over 75% of AGB values being less than 100 g·m^−2^. In summary, the AGB model accurately predicted AGB values and provided a reliable estimate of grassland AGB in Xinjiang.

### 2.2. Spatial and Temporal Dynamic Distribution of Grassland AGB in Xinjiang

#### 2.2.1. Descriptive Statistics of Measured Grassland AGB

From 2000 to 2022, the annual mean values of AGB of different grassland types in Xinjiang ranged from 68.09 to 137.21 g·m^−2^ (Table 1). Among the three major grassland types, the meadow AGB was the highest, followed by steppe AGB, and then desert AGB. Furthermore, according to the grassland area, the meadow makes up 23% of the total grassland area, with the lowland meadow class accounting for the highest percentage (50.42%), whereas the mean montane meadow AGB was the highest (137.21 g·m^−2^). The steppe accounted for 34% of the total grassland area in Xinjiang, with the temperate desert steppe (37.60%) and alpine steppe (31.91%) having relatively large areas, whereas the temperate meadow steppe AGB had the largest mean value (119.56 g·m^−2^). The desert in Xinjiang had the most widespread distribution, accounting for 43% of the grassland area. The area was dominated by the temperate desert (78.00%), with the largest mean AGB value being in the temperate steppe desert (95.51 g·m^−2^), and the smallest AGB was the alpine desert (68.09 g·m^−2^). Additionally, the alpine desert with the smallest grassland area (1.17%) had the lowest mean AGB of 68.09 g·m^−2^, with a standard deviation of 2.81 g·m^−2^ and Coefficient variable (CV) of 0.04, indicating that its AGB was less volatile and less dispersed. The temperate desert with the largest grassland area (33.54%) had an AGB of 88.19 g·m^−2^, with the highest CV (0.05), indicating greater dispersion of temperate desert AGB.

#### 2.2.2. Temporal Changes in Grassland AGB in Xinjiang from 2000 to 2022

The temporal distribution of the mean grassland AGB in Xinjiang from 2000 to 2022 is shown in Figure 3. The AGB of six grassland types, including lowland meadow, temperate steppe desert, temperate desert, alpine desert, temperate desert steppe, and alpine steppe, increased significantly in Xinjiang, with lowland meadow AGB increasing significantly and at the fastest rate (θ = 0.65, *p* < 0.01), followed by alpine steppe AGB and temperate desert AGB (θ = 0.37, *p* < 0.01), and desert grassland AGB (*p* < 0.05). Additionally, montane meadow AGB showed a slight decrease, while other grassland types showed an increasing trend, suggesting that the grassland AGB generally increased from 2000 to 2022.

According to the results of the MK mutation test (Table 2), there was a difference in the timing of AGB mutation. We found that AGB of the alpine grasslands AGB (alpine meadow, alpine steppe, and alpine desert) mutated in 2014, while the AGB of lowland meadow, temperate desert steppe, and temperate steppe desert mutated in 2009. Additionally, temperate steppe and temperate desert had two mutation points, whereas montane meadow and temperate meadow steppe had four mutation points, with the highest CV (0.05) and high volatility.

#### 2.2.3. Spatial Distribution of Grassland AGB in Xinjiang from 2000 to 2022

The spatial distribution of grassland AGB in Xinjiang from 2000 to 2022 is shown in Figure 4a. The high AGB values in the northwestern Yili region, due to its warm climate and abundant rainfall, were suitable for alpine meadows, montane meadows, and temperate meadows. High AGB values were also distributed in mountainous regions such as the Altai and western Tianshan Mountains. The Junggar Basin was dominated by desert grasslands, and its climate was arid and water-scarce with low AGB. The Tarim Basin, which has a large lowland meadow area, had a high AGB. Furthermore, a high AGB was observed in northern Xinjiang. However, AGB was lower in southeastern Xinjiang. In summary, AGB in Xinjiang showed a spatial distribution of high in the northwest and low in the southeast. Figure 4b shows that high CV values were mainly distributed in the Yili River Valley region in the northwest, the Tarim Basin peripheral area, and the northern part of Xinjiang, indicating that grasslands in regions with higher AGB had greater volatility.

The change in grassland AGB in Xinjiang from 2000 to 2022 is shown in Figure 5. The overall trend of AGB had an increasing trend. The growing trend of grassland AGB accounted for 61.19%, of which the significant growth accounted for 16.15%, mainly in the Tarim Basin’s peripheral area and southeastern Xinjiang. The area of the decreasing trend in grassland AGB was 38.81%, which is located in the northwest of Xinjiang. Figure 5c shows that the trend of non-significant increase accounts for the highest proportion, ranging from 40.29% to 58.51%. Among the different grassland types, the lowland meadow had the highest proportion of increasing trends (79.50%), while the montane meadow accounted for the smallest proportion (46.21%). The proportion of increasing trends in most grassland types exceeded 50%.

### 2.3. Influence of Climatic Factors on Grassland AGB Dynamics

#### 2.3.1. Spatial and Temporal Distribution of Climatic Conditions in Grassland AGB

The mean annual temperature of grasslands in Xinjiang from 2000 to 2022 was 2.98 °C, with higher temperatures mainly distributed in the Junggar Basin and margin of the Tarim Basin (Figure 6a). In addition, most of the area (54.65%) showed a warming trend from 2000 to 2022, with an average annual warming trend mainly between 0 and 0.02 °C·a^−1^ (29.8%), and the average warming trend reaching 0.01 °C·a^−1^ (Figure 6c). In other words, the temperature of the grasslands in Xinjiang showed an increasing trend from 2000 to 2022. In addition, the average precipitation in Xinjiang was 288.14 mm during this period, with higher values in the Yili region and southeast Xinjiang (Figure 6b). The annual average precipitation trend was mainly 0–2 mm·a^−1^ (35.78% of the area), with an average precipitation magnitude of 1.59 mm·a^−1^ (Figure 6d). In total, 74.66% of the area showed an increasing trend, indicating that the precipitation of grassland increased from 2000 to 2022. The climate was characterized by warming and humidification. Importantly, warming and humidification trends positively impacted the growth of grasslands in Xinjiang.

#### 2.3.2. Correlation between Climatic Factors and Grassland AGB

The correlation between grassland AGB and temperature from 2000 to 2022 is shown in Figure 7. There was a positive correlation between Xinjiang grassland AGB and temperature, accounting for 52.33% of the total correlation. A significant positive correlation accounted for 3.09% and was mainly distributed in the Altay region in northern Xinjiang and the Hami region in eastern Xinjiang. Furthermore, different grassland types exhibited varying proportions of correlation (Figure 8a), with temperate steppe deserts having the highest positive correlation (62.25%) and highest significant positive correlation (5.76%) and alpine deserts having the lowest proportion (36.96%). The study also showed that the alpine desert had the highest proportion of negative correlations (63.04%), whereas the temperate steppe desert had the lowest (37.75%). In conclusion, there were spatial differences in the correlations between different grassland types and temperature, and the AGB demonstrated a positive correlation with temperature.

There was a positive correlation between grassland AGB and precipitation (Figure 7b,d), accounting for 54.35% of the total area, with a significant positive correlation accounting for 8.38%, mainly found in the Yili River Valley region in western Xinjiang and the northern and eastern parts of Xinjiang. On the other hand, negative correlation accounted for 45.65% of the area, with only 4.96% being significantly negatively correlated, particularly in the northern part of Xinjiang. Significance testing for different grassland types revealed that the alpine meadows had the highest proportion of positive correlation (69.61%), whereas lowland meadows had the highest proportion of negative correlations (61.72%). Additionally, temperate desert grasslands (17.36%) and lowland meadows (3.85%) had the highest and lowest proportions of significant positive correlations with precipitation, respectively (Figure 8b). Overall, the different types of grasslands in Xinjiang were positively correlated with precipitation. From 2000 to 2022, the correlation between grassland AGB and precipitation was greater than that with temperature.

## 3. Discussion

### 3.1. Comparison of AGB Estimates for Different Grasslands

Xinjiang has a variety of grasslands distributed across different regions. Previous studies have shown that the productivity and coverage of grasslands in Xinjiang generally follow the order meadow > grassland > desert [31,32,33,34,35]. Owing to limited ground sample data and significant vertical differences in Xinjiang’s grasslands, there is considerable uncertainty in estimating the grassland AGB in the large region of Xinjiang using remote sensing modeling. Therefore, based on a large number of field observation sample data, this study used an AGB model based on a random forest algorithm to invert the mean value of grassland AGB in Xinjiang to 97.73 g·m^−2^. The mean AGB of desert grassland was 82.27 g·m^−2^ in this study, which was lower than the mean AGB calculated by Fei et al. [36] in Northern Xinjiang (115.42 g·m^−2^). These differences may be due to the location of the sampling sites, the choice of estimation models, or sources of modeling variation. Furthermore, the grassland AGB was the highest in the meadow, second highest in the steppe, and lowest in the desert [37], which is consistent with the results of this study. This study found that the highest value of AGB from 2000 to 2022 was in the montane meadow (152.77 g·m^−2^) in 2016, while the lowest value was in the alpine desert (63.50 g·m^−2^) in 2003.

This study examined the spatial distribution of grassland AGB in Xinjiang. It found that the mean values of grassland AGB in the Yili Valley, Altay Mountains, and western Tianshan Mountains were high and showed an increasing trend, whereas that of the temperate desert grassland AGB in the Junggar Basin was low and showed a decreasing trend over a large area. Chen et al. [38] noted an increasing trend in grassland productivity in the Altay Mountains and Tianshan Mountains but a decreasing trend in the Junggar Basin, which was similar to the results of this study. Furthermore, the grassland AGB in Xinjiang showed spatial distribution characteristics of being high in the northwest and low in the southeast. The overall trend was upward, which was similar to the results of Zhang et al. [39] and Wu et al. [40].

### 3.2. The Main Reasons for the Improvement in AGB in Xinjiang Grassland

The overall grassland AGB has shown a fluctuating upward trend in Xinjiang from 2000 to 2022, which can be explained by the following factors: (1) after 2000, the climate in Xinjiang showed a warm and humid trend [41,42,43], and the increase in temperature and precipitation promoted the growth of grassland; (2) China implemented the “returning pasture to grassland” project and fully implemented the grassland ecological protection subsidy and reward mechanism, reversing the continuous deterioration of grassland without affecting the economic well-being of herders [44]; and (3) Xinjiang has implemented measures such as a grazing ban, grazing rest, zoning rotational grazing, and the establishment of protected areas to reduce the impact of human disturbance on grasslands [45,46,47].

In addition, measurement of the impacts of climate change and human actions on ecological parameters, such as grassland biomass and grassland productivity, showed that climate change had a relatively limited impact on changes in vegetation indicators in the arid and semi-arid regions of Xinjiang. Human activities have greatly impacted the ecosystem processes in Xinjiang’s arid and semi-arid zones [48]. In conclusion, the overall trend of grassland AGB in Xinjiang showed a fluctuating upward trend from 2000 to 2022. A series of grassland resource protection measures carried out in Xinjiang in recent years have been effective, leading to a significant improvement in the ecological environment of the grassland.

### 3.3. Limitations and Future Prospects

Compared to univariate and multivariate regression models, the random forest algorithm demonstrates higher robustness and predictive ability. It is better equipped to handle nonlinear problems and accurately depict the relationship between grassland biomass and modeling variables [49,50]. However, these machine learning models have some limitations and uncertainties. The random forest algorithm prediction is the average of all decision trees, which tends to converge to the mean of the training data, leading to an overestimation or underestimation of AGB [51]. The physical mechanisms of the random forest algorithm for estimating AGB are unclear, but ecosystem models and physical models can better explain the spatiotemporal variations in AGB [52]. Furthermore, the fragmentation of altitude and grassland type also affects the layout of sampling points and impacts the accuracy of the estimation model [53]. Xinjiang’s grassland AGB exhibits significant spatial variation owing to its intricate topographical and climatic conditions. This study considered seven variables and future research should incorporate additional indices such as vegetation index variables [50,53] and climate variables [54] to improve the accuracy of the model.

To better understand the distribution and growth conditions of different grassland types in Xinjiang, it is crucial to continue collecting AGB data and increase the sample size. A combination of machine learning, ecosystem, and physical models should be explored, and more drivers should be screened for different grassland types to develop high-precision AGB inversion models. This will help generate more accurate spatiotemporal distribution maps of AGB in Xinjiang, which is crucial for long-term monitoring and inventory of grassland resources.

## 4. Materials and Methods

### 4.1. Study Area

Xinjiang is located in the central Eurasian continent and on the northwest border of China (Figure 9). It is a temperate continental climate with an arid climate and rare annual precipitation, which occurs mainly from June to August. Precipitation varies greatly in different regions, with the north having higher precipitation than the south, the west having higher precipitation than the east, and the mountains having higher precipitation than the plains. Furthermore, Xinjiang experiences hot summers with small temperature differences between the north and south and cold winters with large temperature differences between the north and south. Xinjiang has complex topographical features, including the Altai Mountains, Junggar Basin, Tianshan Mountains, Tarim Basin, and the Kunlun Mountains, from north to south. The vast area, complex climatic conditions, and topography of Xinjiang have created rich grassland. There are ten types of grasslands in the mountainous and plain areas of Xinjiang, including lowland meadow (11.60%), montane meadow (5.39%), alpine meadow (6.01%), temperate steppe (8.21%), temperate desert steppe (12.78%), temperate meadow steppe (2.16%), alpine steppe (10.85%) temperate steppe desert (8.29%), temperate desert (33.54%), and alpine desert (1.17%).

### 4.2. Data Sources

#### 4.2.1. Sample Data

Grassland biomass data were collected mainly from field surveys during the growing season over the past ten years. The sample points were deployed in a representative and random manner, following the classification principles of grassland types, topographic and geomorphological differences, and the selection of sample plots with flat terrain and typical vegetation types based on the spatial distribution characteristics of different grassland types. Each sample plot was 100 × 100 m, and a small sample square (1 m × 1 m) was placed in each of its four corners and center positions, resulting in five sample squares. The latitude, longitude, elevation, grassland type, and fresh weight at each sampling point were recorded. AGB of grassland was measured and weighed using the flush mowing method, which included collecting samples and drying them in the laboratory at 65 °C for 48 h to constant weight before measuring their dry weights (accuracy of 0.01 g).

#### 4.2.2. Remote Sensing and Climate Dataset

Grassland types were used to classify and estimate the AGB of natural grasslands in Xinjiang. The grassland-type data were based on the 1995 1:1 million grassland-type map of the National Earth System Science Data Center, which was obtained using MSS and TM satellite remote sensing data from the mid-1980s and early 1990s, as well as various geographic data sources.

Seven variables, including temperature (TMP), precipitation (PRE), normalized difference vegetation index (NDVI), enhanced vegetation index (EVI), gross primary productivity (GPP), leaf area index (LAI), and evapotranspiration (ET), were used to construct a random forest algorithm for grassland types in combination with measured data. The meteorological and digital elevation model (DEM) data were collected from the Data Center for Resources and Environmental Sciences at the Chinese Academy of Sciences. MODIS-NDVI, MODIS-EVI, MODIS-GPP, MODIS-LAI, and MODIS-ET were obtained from the National Aeronautics and Space Administration. The data acquisition time was the same as the collection time for the sample plot data, which was unified in July of the same year. The data sources are listed in Table 3.

Temperature and precipitation data from 105 meteorological stations in Xinjiang with complete references and completeness were selected based on the requirements of the study area. Additionally, temperature and precipitation data were interpolated using ANUSPLIN4.4 to obtain meteorological remote sensing data with a spatial resolution of 1 km, which were the driving factors.

### 4.3. Methods

#### 4.3.1. The AGB Model Based on Random Forest Algorithm

Random forest algorithm is a supervised machine learning algorithm based on a decision tree proposed by Breiman in 2001 [55]. It improves the prediction accuracy of the model by summarizing a large number of decision trees and is a new algorithm characterized by rapid calculation and high accuracy that may replace traditional machine learning methods such as neural networks. Especially for dealing with big data and a large number of variables, the effect of a large number of variables can be predicted very well, and it is known as one of the best algorithms at present [56].

The prediction model based on the random forest algorithm is a branch of the machine learning model, which is a kind of integrated model. It is an algorithm that integrates multiple trees via the idea of integrated learning, and its basic unit is the decision tree. There are two important concepts involved here: decision tree and ensemble learning. A decision tree is a tree-like structure where each internal node represents a test on an attribute, each branch represents a test output, and each leaf node represents a class. In a word, a decision tree is a series of judgments to classify data based on their characteristics. Ensemble learning solves a single prediction problem by building a combination of several models. It works by generating multiple models that learn and make predictions independently and form a final prediction that is better than any single model [57].

In general, it builds a forest of many decision trees by randomly generating hundreds to thousands of decision trees and then selects the tree with the highest degree of repetition as the final result (Figure 10a). The following are six steps for constructing a regression prediction model using a random forest algorithm:

(1) Collect data sets: Collect data sets that will be used for training in the random forest model. 

(2) Random sampling: Multiple sub-data sets are randomly extracted from the training data set. These sub-data sets are usually the same size as the original data set, but their samples are randomly selected. 

(3) Construct a decision tree: For each sub-data set, a decision tree is built independently, and all the data run along the tree. All of the data are run down the tree, and proximities are computed for each pair of cases.

(4) Integrate decision tree: Using the constructed decision tree to form a random forest. For regression problems, the average method is usually used, that is, the average of the regression results of multiple trees. 

(5) Predict and evaluate its performance: Use a constructed random forest model to make regression predictions. Validation data sets can be used to evaluate the performance of the model.

(6) Adjust the parameters and apply the model: According to the results of the model performance evaluation, the parameters are adjusted to optimize the performance of the model and applied to the new data for regression prediction.

NDVI, EVI, GPP, ET, LAI, TMP, and PRE were selected as independent variables to build the model in this paper (Figure 10b). The measured grassland AGB was used as the dependent variable. They belong to the step of collecting the dataset. In addition, the random forest algorithm was built using the Scikit-learn tool in Python3.9, and the importance of each independent variable was evaluated. They belong to the step of constructing and integrating decision trees. Machine learning methods typically require cross-validation to detect overfitting [58]. The optimal number of inputs was determined using a k-fold cross-validation. This study employed 10-fold cross-validation to ensure that each subgroup participated in both training and testing, thereby reducing the generalization error. Furthermore, this step involves evaluating the model performance and tuning the parameters to ensure the optimized model can be applied to regression prediction.

#### 4.3.2. Model Validation

We randomly selected 80% of the measured samples for modeling analysis and 20% for model verification. This study used the coefficient of determination (R^2^), root mean square error (RMSE), and mean absolute error (MAE) to assess model accuracy. R^2^ indicates the proximity of the observed values to the fitted regression line or the proportion of variance explained by the predictors. RMSE calculates the error between predicted and measured values. MAE calculates the average absolute differences between predicted and measured values. The closer R^2^ is to 1, and the smaller the RMSE and MAE, the stronger the model’s estimation ability and the higher its accuracy and stability. The calculation formula for *MAE* is as follows:(1)$$MAE=\frac{1}{N}\sum_{i=1}^{N}\left|Y_{i}-{Y_{i}}^{\prime }\right|$$
where $Y_{i}$ is the measured grassland AGB. ${Y_{i}}^{\prime }$ is the simulated grassland AGB. $\bar{Y}$ is the average value of measured grassland AGB. N is the total number of sample points.

#### 4.3.3. Trend Analysis

The temporal trend of the estimated AGB dataset was analyzed using univariate linear regression to calculate the AGB change rate for each pixel, which reflects the overall spatial change pattern. The formula is
(2)$$Slope=\frac{n\times \sum_{i=1}^{n}i\times x_{i}-\left(\sum_{i=1}^{n}x_{i}\right)\left(\sum_{i=1}^{n}i\right)}{n\times \sum_{i=1}^{n}i^{2}-{\left(\sum_{i=1}^{n}i\right)}^{2}}$$
where $x_{i}$ is the AGB value in year i. n is the number of years, and i ranges from 1 to 23. *Slope* is the slope of the regression equation. When the *slope* is greater than 0, the AGB of the grassland shows an increasing trend and vice versa.

#### 4.3.4. Correlation Analysis

To study the grassland AGB in Xinjiang and its correlation with temperature and precipitation factors, we used the following equation:(3)$$R_{xy}=\frac{\sum_{i=1}^{n}\left(x_{i}-\bar{x}\right)\left(y_{i}-\bar{y}\right)}{\sqrt{\sum_{i=1}^{n}{\left(x_{i}-\bar{x}\right)}^{2}\sum_{i=1}^{n}{\left(y_{i}-\bar{y}\right)}^{2}}}$$
where $R_{xy}$ is the correlation coefficient of variables x and y; $x_{i}$ is the AGB value in year i, $y_{i}$ is the value of temperature and precipitation in year i. $\bar{x}$ and $\bar{y}$ are the mean values of temperature and precipitation, respectively.

#### 4.3.5. Mann–Kendall Mutation Test

The Mann–Kendall test is a non-parametric statistical test. For mutation testing of time series (x), we need to define a statistic. The calculation equation was as follows:(4)$$S_{k}=\sum_{i=1}^{k}\sum_{j}^{i=1}M_{ij},(k=2,3,\ldots ,n)$$
where *n* is the sample size. When $x_{i}>x_{j}$, $M_{ij}=1$; when $x_{i}<x_{j}$, $M_{ij}=0$, (*j* = 1, 2, ⋯, *i*).

Assuming that the time series are randomly independent, we define the statistics:(5)$$UF_{k}=\frac{S_{k}-{\bar{S}}_{k}}{\sqrt{Var(S_{k})}},(k=1,2,\ldots ,n)$$
where ${\bar{S}}_{k}=k(k+1)/4$; $Var(S_{k})=k(k-1)(2k+5)/72$.

$UF_{k}$ is the standard normal distribution, which is a sequence of statistics calculated in the order of the time series. We are then given the significance level a. If $\left|UF_{k}\right|>U_{\alpha }$, there is a significant change in trend. We sort the time series in reverse order and repeat the above equation and make $UB_{k}=-UF_{k}$, (k=n,n−1,n−2,…,2,1), and $UB_{1}=0$. When $UF_{k}>0$, the series is on an upward trend; when $UF_{k}<0$, the series is on a downward trend, and exceeding the threshold indicates a clear trend in the series. The intersection of the two curves, $UF_{k}$ and $UB_{k}$, is the point at which the mutation begins.

## 5. Conclusions

In this study, we developed a suitable AGB model for Xinjiang grasslands based on the random forest algorithm, using AGB observation data, remote sensing vegetation indices, and meteorological data. We analyzed the grassland AGB spatiotemporal changes and further explored the response of AGB to variations in climatic factors. The main conclusions are as follows:

The grassland AGB model based on the random forest algorithm fit well, with a coefficient of determination and RMSE of 0.55 and 64.33 g·m^−2^, respectively. The estimated values of the model were close to the actual values, making them suitable for monitoring changes in grassland AGB.

From 2000 to 2022, the spatial distribution of grassland AGB in Xinjiang showed high levels in the northwest and low values in the southeast, with an overall upward trend. All grassland types, except montane meadows, showed increasing trends, with lowland meadows showing the fastest growth rate (0.65 g·m^−2^·a^−1^). Additionally, 16.15% of the total area showed a significant increase in AGB, mainly located in the periphery of the Tarim Basin and southeastern part of Xinjiang.

Xinjiang’s climate is experiencing warming and increased humidity, leading to a positive impact on AGB growth due to increased precipitation and warming. Furthermore, temperature and precipitation were positively correlated with AGB, but precipitation correlates better with AGB than temperature.

## Figures and Tables

**Figure 1 plants-13-00548-f001:**
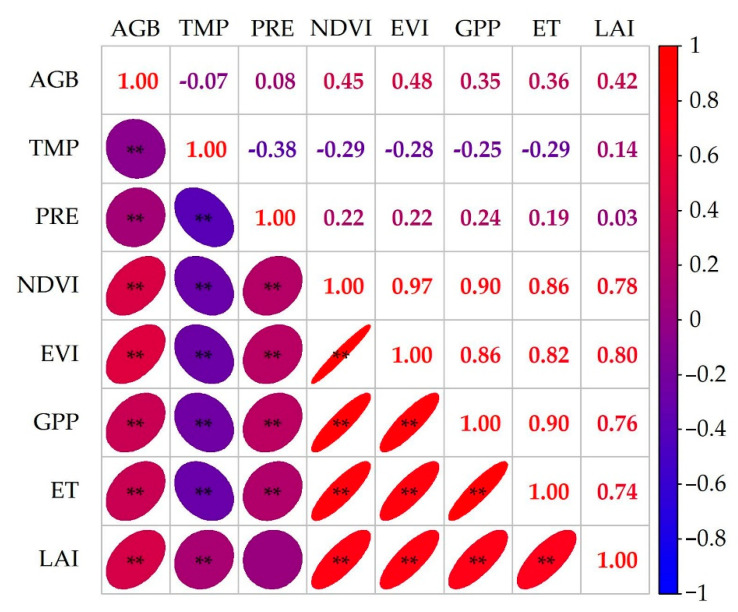
Correlation between AGB and various environmental variables (** represent significance at the levels of 1%).

**Figure 2 plants-13-00548-f002:**
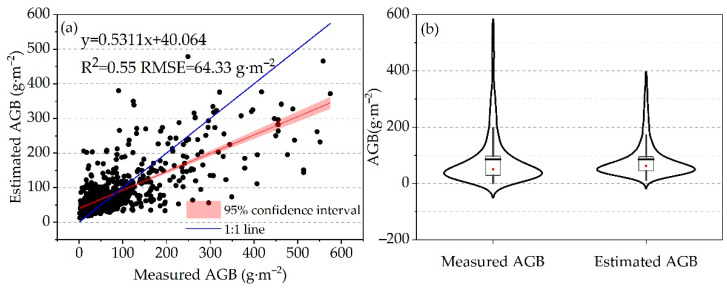
The relationship between the measured grassland AGB and the estimated grassland AGB in Xinjiang. (**a**) Linear correlation. (**b**) Violin plot.

**Figure 3 plants-13-00548-f003:**
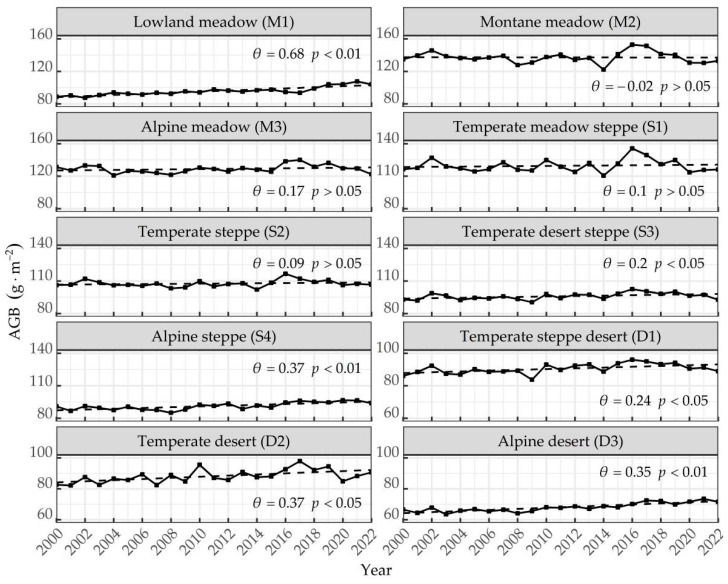
Characteristics of temporal changes in different grassland AGB from 2000 to 2022.

**Figure 4 plants-13-00548-f004:**
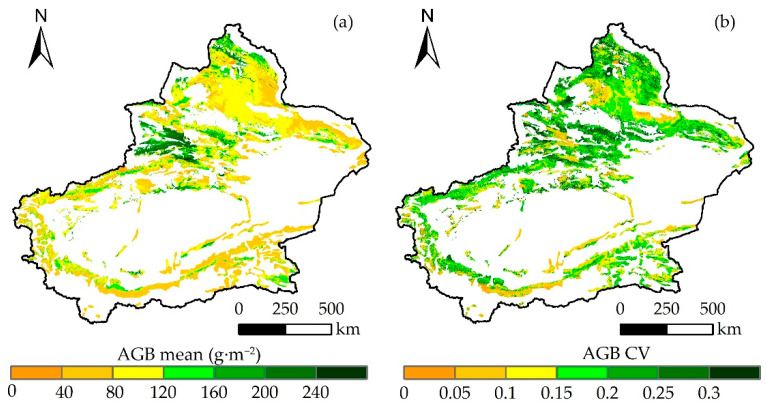
Spatial distribution of the mean (**a**) and coefficient of variation (**b**) of grassland AGB from 2000 to 2022.

**Figure 5 plants-13-00548-f005:**
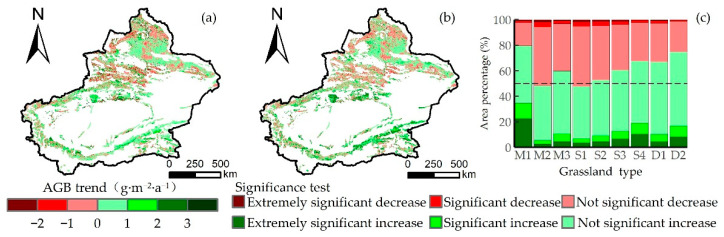
(**a**,**b**) Spatial distribution of grassland AGB trends and significance tests in Xinjiang from 2000 to 2022. (**c**) Statistical data on AGB trends for different grassland types.

**Figure 6 plants-13-00548-f006:**
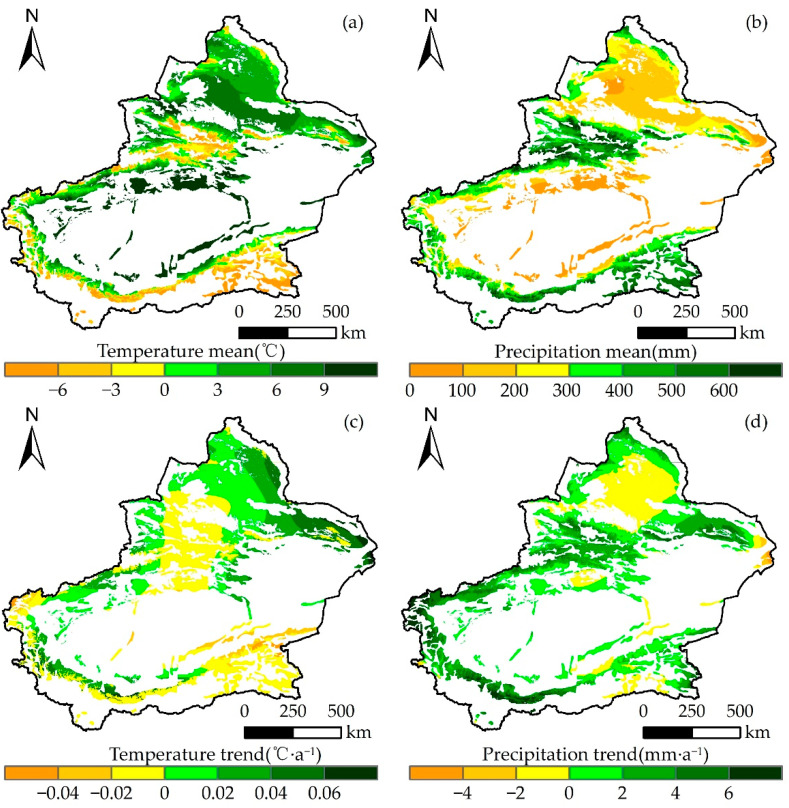
Magnitude and trend in mean annual temperature and mean annual precipitation in Xinjiang over the period 2000–2022. (**a**) Average annual temperature (°C); (**b**) average annual precipitation (mm); (**c**) the trend of temperature (°C·a^−1^); (**d**) the trend of precipitation (°C·a^−1^).

**Figure 7 plants-13-00548-f007:**
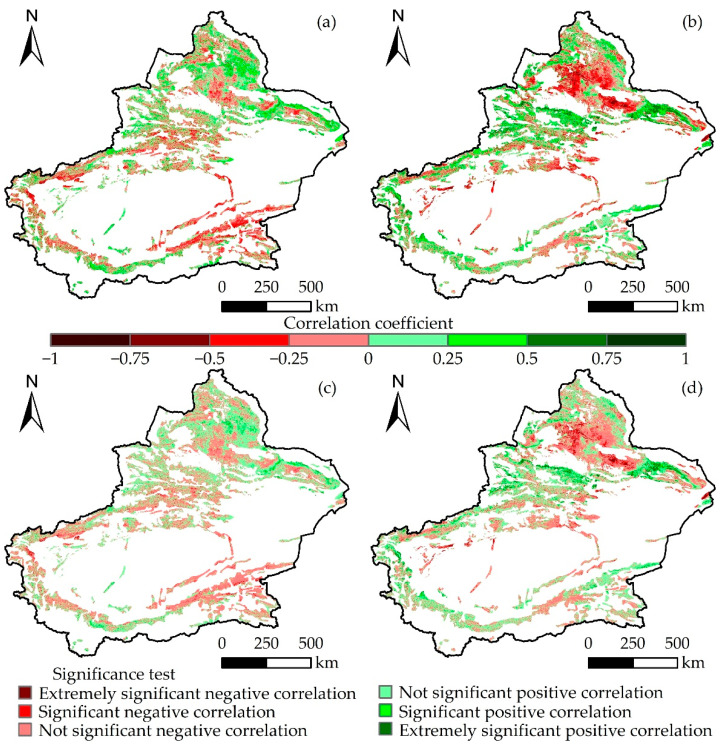
Spatial distribution of correlation and significance test between AGB and meteorological factors in Xinjiang from 2000 to 2022: (**a**,**c**) temperature and (**b**,**d**) precipitation.

**Figure 8 plants-13-00548-f008:**
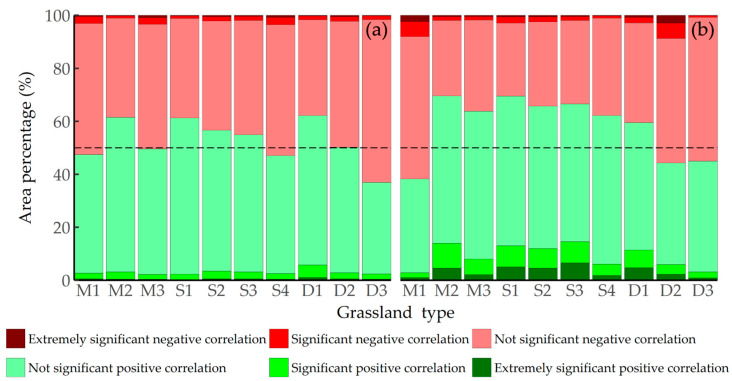
Statistics data of correlation between biomass and climate factors for different grassland types: (**a**) AGB and temperature and (**b**) AGB and precipitation.

**Figure 9 plants-13-00548-f009:**
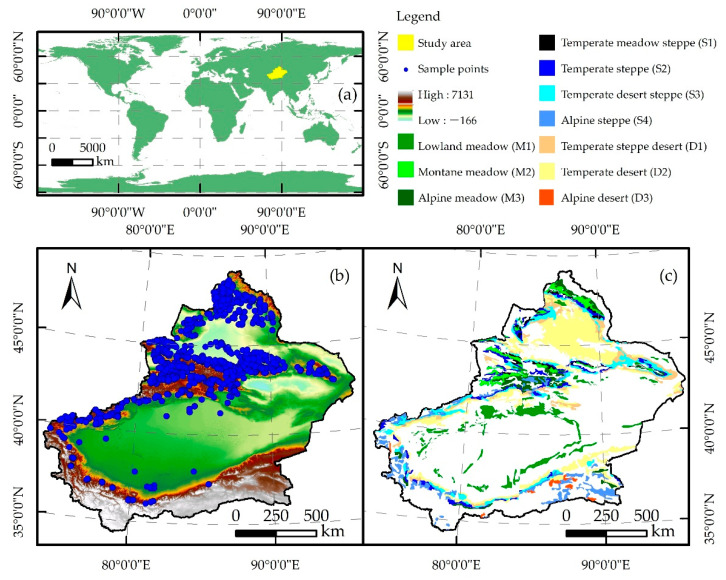
Geographic location (**a**), sampling point distribution (**b**), and grassland-type map (**c**) on Xinjiang, China.

**Figure 10 plants-13-00548-f010:**
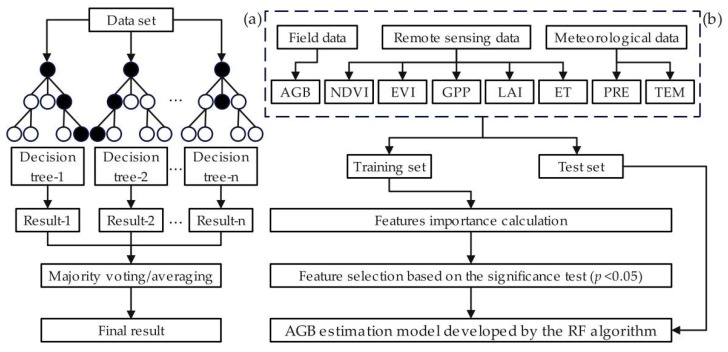
(**a**) Flowchart of the random forest algorithm. (**b**) Flowchart of the grassland AGB estimation.

**Table 1 plants-13-00548-t001:** Descriptive statistics of grassland AGB in Xinjiang from 2000 to 2022.

Grassland Types	Mean Value (g·m^−2^)	Maximum Value (g·m^−2^)	Minimum Value (g·m^−2^)	Standard Deviation (g·m^−2^)	Coefficient Variable
Lowland meadow (M1)	95.69	107.59	87.50	5.20	0.05
Montane meadow (M2)	137.21	152.77	122.18	6.94	0.05
Alpine meadow (M3)	128.91	140.02	120.89	4.98	0.04
Meadow (M)	120.60	152.77	87.50	18.95	0.16
Temperate meadow steppe (S1)	119.56	135.62	110.52	5.86	0.05
Temperate steppe (S2)	107.64	116.65	102.00	3.20	0.03
Temperate desert steppe (S3)	95.91	102.49	90.41	3.08	0.03
Alpine steppe (S4)	91.33	96.59	85.10	3.37	0.04
Steppe (S)	103.61	135.62	85.10	11.72	0.11
Temperate steppe desert (D1)	90.51	96.03	83.76	3.08	0.03
Temperate desert (D2)	88.19	97.90	82.17	4.32	0.05
Alpine desert (D3)	68.09	73.50	63.51	2.81	0.04
Desert (D)	82.27	97.90	63.51	10.70	0.13

**Table 2 plants-13-00548-t002:** Mutation time of grassland AGB.

Grassland Types	Year of Mutation
Lowland meadow (M1)	2009
Montane meadow (M2)	2004/2010/2014/2022
Alpine meadow (M3)	2014
Temperate meadow steppe (S1)	2005/2007/2010/2013
Temperate steppe (S2)	2003/2014
Temperate desert steppe (S3)	2009
Alpine steppe (S4)	2014
Temperate steppe desert (D1)	2009
Temperate desert (D2)	2006/2009
Alpine desert (D3)	2014

**Table 3 plants-13-00548-t003:** Introduction to data sources.

Data	Year	Resolution	The Data Source
Grassland data	1995	/	http://www.geodata.cn/ (accessed on 10 March 2022)
Temperature	2000–2022	1 km	http://www.resdc.cn (accessed on 5 April 2022)
Precipitation	2000–2022	1 km	http://www.resdc.cn (accessed on 10 April 2022)
DEM	1995	90 m	http://www.resdc.cn (accessed on 20 May 2022)
MODIS-NDVI	2000–2022	500 m	https://ladsweb.modaps.eosdis.nasa.gov/ (accessed on 12 May 2023)
MODIS-EVI	2000–2022	500 m	https://ladwedb.modaps.eosdis.nasa.gov/ (accessed on 19 May 2023)
MODIS-GPP	2000–2022	500 m	https://ladwedb.modaps.eosdis.nasa.gov/ (accessed on 12 June 2023)
MODIS-LAI	2000–2022	500 m	https://ladwedb.modaps.eosdis.nasa.gov/ (accessed on 8 July 2023)
MODIS-ET	2000–2022	500 m	https://ladwedb.modaps.eosdis.nasa.gov/ (accessed on 20 July 2023)

## Data Availability

Dataset available on request from the authors.

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
