# Peer review of "The Estimation of Grassland Aboveground Biomass and Analysis of Its Response to Climatic Factors Using a Random Forest Algorithm in Xinjiang, China"

_plants, 2024, doi:10.3390/plants13040548_

Round 1
Reviewer 1 Report
Comments and Suggestions for Authors
Non needed.
Comments on the Quality of English LanguageThere are minor grammatical errors, for example, line 38 should read "Xinijang has abundant grasslands..." not abundance.
Reviewer 2 Report
Comments and Suggestions for Authors
"The document is very comprehensive. The sections of the document are accurate, and there is a good balance in their length. The data analysis for obtaining the results, as well as its discussion, is thorough and coherent. Additionally, in the discussion section, it is mentioned that the model can be improved by adding new variables, implying ongoing enhancements. From my perspective, I would have liked a future simulation considering climate change to assess the potential coherence of the study, but I suppose this could lead to another possible paper. In my view, it is ready for publication."
Reviewer 3 Report
Comments and Suggestions for Authors
In the Xinjiang region there are four meadow types (23%) , four steppe types (34%) and two desert types (43%). From the total surface of of grassland in the area the temperate desert represents 33.54%. A random forest estimation model was applied for a time span of 22 years in order to estimate spatial distribution and and trend for all grassland types in Xinjiang.
The aboveground biomass (AGB) is considered very important for long-term monitoring and inventory of grassland resources. The study revealed an improvement of AGB correlated with the trend of climate to become more warm and humid, the implementation of the project "returning pasture to graassland" and implementation of some measures as grazing ban, grazing rest, zoning rotational grazing, establishment of protected areas.
There were made correlations with temperature and precipitations and were used some indices in combination with measured data in the field.
I recommend the authors to develop in another paper the importance of biodiversity value of the meadows, especially for desert habitats.
Reviewer 4 Report
Comments and Suggestions for Authors
The authors present an interesting study on the spatiotemporal variation of ABG and its response to climatic factors. Using RFM, the authors analyze the changes that occurred between 2000 and 2022 in the Xinjiang region (China), where environmental conditions are harsh. This type of study is interesting because the changes occurred on grasslands, beyond ecological knowledge, affects the ecosystem services provided by them.
The research is well done, the methodology fits perfectly with the purpose of the study, and the discussion and conclusions support the results obtained. The graphs and tables support and clarify the main results.
However, I'd like to suggest some minor changes for final approval.
General aspects:
1. I don't know why, but along the text there are several words with an interspersed hyphen (e.g. line 50 page 2: im-prove). Check the text carefully and correct this.
2. In general, the captions should be a little more detailed. For example, in Figure 1 (page 3), you should add .....-type map on the Xinjiang region (China).
Detailed aspects:
1. Title. I'll add this "...climatic factors in the Xinjiang region (China)..." to give readers the geographical context.
2. The first time Xinjiang appears in the regular text (line 38 of page 1), I'll add the reference to Xinjiang Region (China).
3. Line 63 page 2, better "popular" that mainstream.
4. Figure 1. A global (small) map of the region (Central Asia) is needed to give the readers the geographical context.
5. Sentences on lines 280-281 (page 10) should be rewritten because they repeat the same thing, maybe the two sentences could be combined into one.
6. Figure 9 (page 13). Better "not significant" than "insignificant".
7. Sentence between lines 396 and 399 (page 14) should be rewritten for better understanding.
8. Line 418 (page 14), the term ecosystem is orphaned in the text. Please check it.
As you can see, only minor considerations. I've enjoyed reading the research. Congratulations.
Comments on the Quality of English Language
